# Systematic identification of genes encoding cell surface and secreted proteins that are essential for *in vitro* growth and infection in *Leishmania donovani*

**Adam J. Roberts**[1], **Han B. Ong**[1], **Simon Clare**[2], **Cordelia Brandt**[2], **Katherine Harcourt**[2], **Susanne U. Franssen**[3], **James A. Cotton**[3], **Nicole Müller-Sienerth**[1], **Gavin J. Wright**[1,4]*

**1** Cell Surface Signalling Laboratory, Wellcome Sanger Institute, Hinxton, Cambridge, United Kingdom, **2** Pathogen Support Team, Wellcome Sanger Institute, Hinxton, Cambridge, United Kingdom, **3** Parasite Genomics, Wellcome Sanger Institute, Hinxton, Cambridge, United Kingdom, **4** Department of Biology, Hull York Medical School, York Biomedical Research Institute, University of York, York, United Kingdom

* gavin.wright@york.ac.uk

**Data Availability Statement:** All relevant data are within the manuscript and its Supporting Information files.

## Abstract

Leishmaniasis is an infectious disease caused by protozoan parasites belonging to the genus *Leishmania* for which there are no approved human vaccines. Infections localise to different tissues in a species-specific manner with the visceral form of the disease caused by *Leishmania donovani* and *L. infantum* being the most deadly in humans. Although *Leishmania* spp. parasites are predominantly intracellular, the visceral disease can be prevented in dogs by vaccinating with a complex mixture of secreted products from cultures of *L. infantum* promastigotes. With the logic that extracellular parasite proteins make good subunit vaccine candidates because they are directly accessible to vaccine-elicited host antibodies, here we attempt to discover proteins that are essential for *in vitro* growth and host infection with the goal of identifying subunit vaccine candidates. Using an *in silico* analysis of the *Leishmania donovani* genome, we identified 92 genes encoding proteins that are predicted to be secreted or externally anchored to the parasite membrane by a single transmembrane region or a GPI anchor. By selecting a transgenic *L. donovani* parasite that expresses both luciferase and the Cas9 nuclease, we systematically attempted to target all 92 genes by CRISPR genome editing and identified four that were required for *in vitro* growth. For fifty-five genes, we infected cohorts of mice with each mutant parasite and by longitudinally quantifying parasitaemia with bioluminescent imaging, showed that nine genes had evidence of an attenuated infection although all ultimately established an infection. Finally, we expressed two genes as full-length soluble recombinant proteins and tested them as subunit vaccine candidates in a murine preclinical infection model. Both proteins elicited significant levels of protection against the uncontrolled development of a splenic infection warranting further investigation as subunit vaccine candidates against this deadly infectious tropical disease.

**Funding:** This work was funded by the Wellcome Trust https://wellcome.org/ (grant 206194) awarded to GJW. The funders had no role in study design, data collection and analysis, decision to publish, or preparation of the manuscript.

**Competing interests:** The authors have declared that no competing interests exist.

## Author summary

Leishmaniasis is a parasitic infectious disease that is responsible for many tens of thousands of human deaths per year, primarily in impoverished parts of the world. Although there are drugs to treat this parasite infection, resistance is emerging and there are no approved human vaccines. Extracellular parasite proteins can make good vaccine targets because they are directly accessible to host antibodies; however, not all parasite surface proteins can elicit protective immune responses. With the goal of identifying new vaccine targets, we selected over 90 genes that encode parasite cell surface and secreted proteins and used the latest CRISPR gene editing technology to individually target them. Using these mutant parasites, we identified four genes required for parasite growth in the laboratory. We expressed two of the proteins as subunit vaccines and a preclinical infection model was used to determine if they could elicit protective immune responses. We found that two of our candidates were able to confer significant levels of protection in a murine model of visceral leishmaniasis. Our study will contribute to the search for a highly effective vaccine against visceral leishmaniasis to improve the lives of people living in some of the poorest regions on the planet.

## Introduction

The neglected tropical infectious diseases that are collectively known as the Leishmaniases are caused by one of over twenty different species of single-celled parasites from the genus *Leishmania*. The clinical manifestations of the infection are divided into distinct categories: a non-fatal cutaneous form characterised by the presence of painful but usually self-healing ulcers on the skin; a mucocutaneous form that is more aggressive and can lead to severe destruction of the mucosal tissues of the mouth and nose; and finally, a visceral form that is fatal if left untreated. This visceral form accounts for approximately twenty thousand deaths per annum, although this is widely considered a limiting lower bound due to the significant levels of underreporting [1]. Although an active area of research [2,3], no new drugs have yet been licenced and so patients must therefore rely on existing pentavalent antimonials, miltefosine and amphotericin B (AmB), which have varying levels of efficacy. Sodium stibogluconate use in Bihar, for example, fails in up to 60% of cases [4], and both treatment failures and increasing relapse rates have been reported for miltefosine [5,6]. The need to treat each new infection, and the concerns around the evolution of parasite drug resistance means that a useful control tool would be the development of an effective vaccine, however, there are no vaccines against *L. donovani* and *L. infantum* that are approved for human use.

Laboratory studies have demonstrated that prior infection with genetically attenuated visceralising *Leishmania* species [7–9], or those causing cutaneous infections [10–12] can protect against subsequent infections with parasites that cause visceral leishmaniasis. While these findings provide encouragement that a vaccine against visceral leishmaniasis could be achieved, live attenuated parasite vaccines are not favoured due to the risk of breakthrough infections, and the challenges and high costs of manufacturing a consistent biological product. Modern vaccines are usually chemically defined and are typically composed of purified recombinant proteins that are easier to manufacture consistently. A major challenge in parasite vaccine development has therefore been to identify those antigens that are capable of eliciting protective immune responses in the host from the several thousand encoded within the parasite genome [13,14]. Although *Leishmania* parasites invade and develop within cells, it is known

that strong protection can be induced in dogs by vaccinating with the concentrated acellular supernatant from *in vitro* cultures of *L. infantum* promastigotes [15], with evidence that humoral immunity plays an important role [16]. This provides some evidence that proteins displayed on the surface or secreted by the parasite could be viable vaccine targets for visceral leishmaniasis. While restricting the search for an effective vaccine candidate to secreted and surface-exposed parasite proteins rationalises the problem to some degree, there are still a large number of possible candidates and not all are capable of eliciting protective immunity.

One approach to identify and prioritise vaccine candidates for visceral leishmaniasis is to determine which proteins are critically required for viability and virulence with the rationale that targeting them with vaccine-induced antibodies will neutralise the parasite. The development of CRISPR/Cas9 gene editing technologies and their application to *Leishmania* [17,18] permits the systematic creation of large libraries of gene-targeted parasites. This, together with the ability of *L. donovani* to infect experimentally-tractable small rodents in preclinical models, and advances in the longitudinal quantification of parasite infections using whole-animal bioluminescent imaging with strains of luciferase-expressing parasites [19–23], make it possible to experimentally determine the protective effect of any putative vaccine *in vivo*.

Here, we have selected a bioluminescent strain of *L. donovani* expressing the *Streptococcus pyogenes* Cas9 protein (SpCas9) that is virulent in a murine infection model and used it to systematically target 92 *Leishmania donovani* genes encoding putative cell surface and secreted proteins. Using this approach, we identified several genes that were required for *in vitro* parasite growth and two recombinant protein subunit vaccine candidates that could elicit protection against the development of the splenic disease.

## Results

### Selection of an infectious Cas9-expressing bioluminescent *L. donovani* cell line

To identify *L. donovani* genes that are predicted to encode cell surface and secreted proteins that are required for either *in vitro* growth or host infection, we first needed to generate a transgenic bioluminescent parasite that expressed both the Cas9 nuclease and T7 RNA polymerase [17,24]. We selected an *L. donovani* LV9 parasite strain stably expressing both the fluorescent protein mCherry and firefly luciferase as a parental cell line to facilitate *in vitro* and *in vivo* characterisation [19]. Parasites were electroporated with a plasmid encoding both SpCas9 and T7 RNA polymerase, and we confirmed that the resulting transgenic parasites expressed the SpCas9 protein (Fig 1A). To demonstrate functional expression of the T7 RNA polymerase and usability of the parasite line for genetic targeting, we transfected the line with four PCR products: two containing templates for sgRNAs driven by a T7 promoter matching the 5' and 3' untranslated regions of the gene encoding the flagellum protein *PF16*, and two encoding genes that confer resistance to the drugs puromycin and blasticidin, each flanked by 30 nucleotides of homology 5' and 3' of the cut sites [17,24]. A parasite population resistant to both puromycin and blasticidin was only observed in parasites electroporated in the presence of the four PCR products, and diagnostic PCR analysis of the genomic DNA confirmed targeted disruption of the *PF16* locus (Fig 1B).

Prolonged *in vitro* culture of *Leishmania spp.* has previously been attributed to a loss of virulence in experimental settings [25,26], and so to ensure the virulence of the line, mice were inoculated and parasites recovered from the spleen of an infected mouse after 37 days. Parasites were cloned by limiting dilution and 12 selected clones were again tested for their ability to target the *PF16* locus as above; the clone that resulted in the most rapid generation of resistance to both drug selection markers was selected. Infection parameters of this cloned line

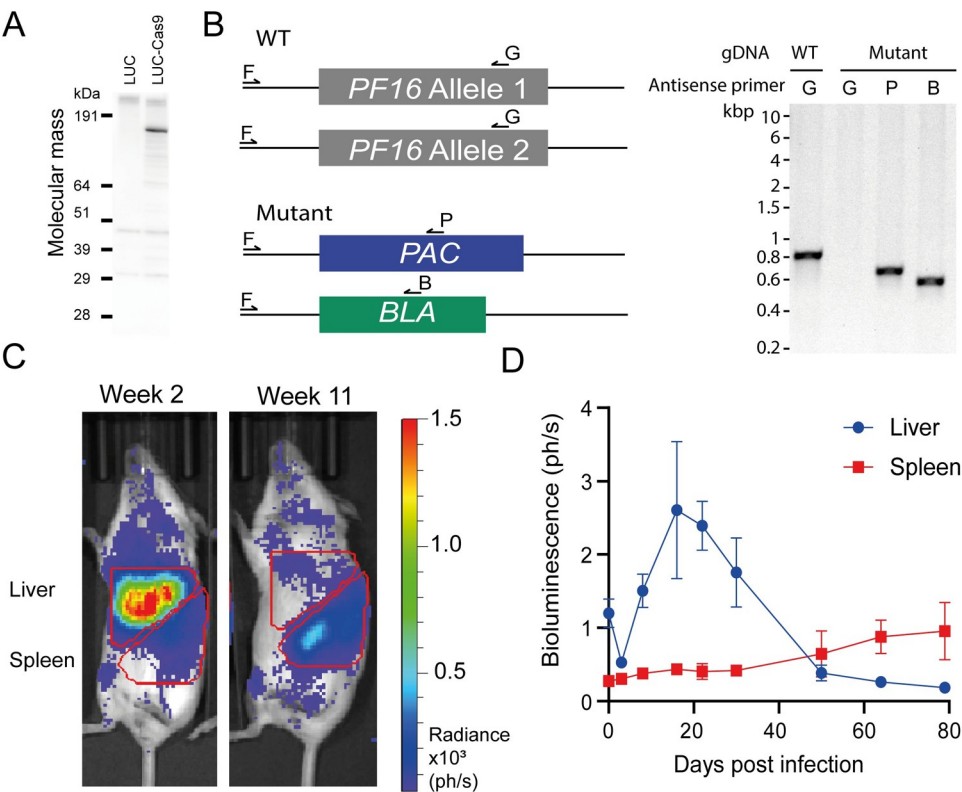

**Fig 1. Selection and characterization of a virulent bioluminescent *L. donovani* cell line expressing SpCas9. (A)** Stable expression of a FLAG-tagged Cas9 nuclease in a bioluminescent *L. donovani* parasite line (LUC). Total cell lysates from 5x10^6 *L. donovani* promastigotes were resolved by SDS-PAGE under reducing conditions, blotted, and detected with an anti-FLAG antibody. (**B**) Schematic of diagnostic PCR to demonstrate expected targeting, gDNA from wild type (WT) or doubly-drug resistant (mutant) parasites is amplified using a shared sense primer (F) and specific antisense primers targeting the gene of interest (G), *PAC* (P) or *BLA* (B) (left panel). Products from a diagnostic PCR demonstrating replacement of the endogenous *PF16* gene in *L. donovani* using Cas9 genome editing. Primers specific for the endogenous *PF16* locus (primers F and G) amplified a product of the expected size from genomic DNA from the parental (lane 1) but not the drug-resistant parasites (lane 2); targeted integration into the *PF16* locus was demonstrated with shared and drug-specific primers (P and B) for the puromycin (lane 3) and blasticidin resistance genes (lane 4) respectively (right panel). (**C**) Representative images of mice infected with Ld-LUC-T7-Cas9 at weeks 2 and 11 post infection including the gating strategy to quantify parasitemia in the liver and spleen. (**D**) Virulence of the Ld-LUC-T7-Cas9 line in an experimental murine infection model. Groups of 3 to 5 female BALB/c mice were infected with stationary phase promastigotes of the Ld-LUC-T7-Cas9 line and parasitemia in both liver and spleen quantified using bioluminescent imaging. Data points represent means ± s.d. Data are from a representative experiment. Each unit of bioluminescence represents 1 x 10^5 photons per second.

were determined using bioluminescence in mice and were shown to infect both the liver and spleen of BALB/c mice at similar levels and kinetics to those observed with the parental strain [19] (Fig 1C and 1D) [19]. This parasite line was named Ld-LUC-T7-Cas9 and used in all subsequent experiments.

## A library of gene-targeted *L. donovani* parasites encoding cell surface and secreted proteins identified four genes required for *in vitro* culture

To identify genes encoding cell surface or secreted proteins that are required for *L. donovani* cell viability *in vitro* and *in vivo*, we used the Ld-LUC-T7-Cas9 line to systematically select gene-deficient parasites from a curated list of 92 putative cell surface and secreted *Leishmania* proteins for which there was evidence of protein expression from published proteomics data;

for convenience, each candidate was given a systematic "LD" number (S1 Table). To increase the chances of correct gene targeting, we sequenced the genome of the Ld-LUC-T7 parental strain and compared it to a reference genome sequence [14] to identify polymorphisms in the predesigned targeting and repair primers, incorporating any differences as appropriate. Of the 92 candidates targeted, doubly-drug resistant parasite populations with confirmed locus-specific biallelic knockout with each of the drug selection markers was confirmed for 68 genes (74%) by PCR (S1 Fig). Typically, parasite strains successfully targeted at both alleles required just one or two attempts, demonstrating that these genes were readily dispensable for *in vitro* promastigote proliferation (S1 Table). Just under a fifth (17/92) of genes demonstrated evidence of genomic rearrangements in at least one targeting attempt because despite the correct incorporation of both drug resistance markers, a PCR product showing the retention of the native locus was also observed, most likely due to selection-induced aneuploidy [27]. There was not an exclusive relationship, however, between the retention of the native locus after targeting, and the chromosomes on which the targeted genes were located, and this was true for chromosomes where multiple genes had been targeted. Furthermore, analysis of the normalised read depth of the sequencing data from the parental strain suggested the presence of trisomy and tetrasomy on chromosomes 26 and 31 respectively in Ld-LUC-T7-Cas9, and both these chromosomes contained genes that we were able to target successfully using only the two selectable markers as evidenced by the disruption of the native loci of LD40, which is present on chromosome 31, and LD30, LD31 and LD73 that are all located on chromosome 26 (Fig 2A and S1 Table). A possible mechanism for this observation is the integration of a selectable marker into multiple alleles. The results of PCR genotyping for three genes (*LdBPK_090870.1* (LD17), *LdBPK_141150.1* (LD21) and *LdBPK_140570.1* (LD55)) were consistently ambiguous, because we were unable to demonstrate the presence of the endogenous locus and so these genes were not further investigated in this study.

The targeting of four genes: *LdBPK_211610.1* (LD11), *LdBPK_100590.1* (LD18), *LdBPK_111030.1* (LD67) and *LdBPK_010280.1* (LD81) failed to recover any viable parasites despite a minimum of five independent attempts (Table 1). To confirm that the targeting and repair constructs for these genes were functioning as expected, we attempted single allele replacement for each gene using just the puromycin selection cassette. For all four of the genes (LD11, LD18, LD67 and LD81), we were able to recover viable puromycin-resistant parasites, validating the specificity of the designed targeting sequences (Fig 2B and 2C).

To demonstrate that these genes were required for *in vitro* parasite viability, we attempted to replace both endogenous alleles in either the presence or absence of a constitutively expressed version of the same gene that lacked the CRISPR targeting sequences. Three independent transfections were performed and doubly drug-resistant parasite populations were recovered for both LD11 and LD81 only in the presence of a non-targetable copy with a 100% success rate demonstrating their essentiality (Fig 2D). PCR genotyping confirmed the correct targeting of both endogenous alleles of LD11 in the presence of the ectopic rescue (Fig 2E); however, we were unable to obtain PCR genotyping for the LD81 parasite populations despite multiple attempts. Viable drug resistant parasites were not recovered from electroporations targeting LD18 or LD67 in either the absence or presence of their appropriate overexpressed open reading frame. This suggested that our failure to replace both endogenous alleles of LD18 and LD67 might be because the genetic rescue was insufficient to support the loss of both endogenous copies, or allelic variation in the targeting regions acquired by the parasite post genome sequencing prevented the replacement of the second allele with the drug resistance marker [28,29].

*Leishmania* parasites exhibit remarkable genomic plasticity, with the parasites being able to tolerate aneuploidy of different chromosomes both *in vitro* and *in vivo* [14,27]. The ability to

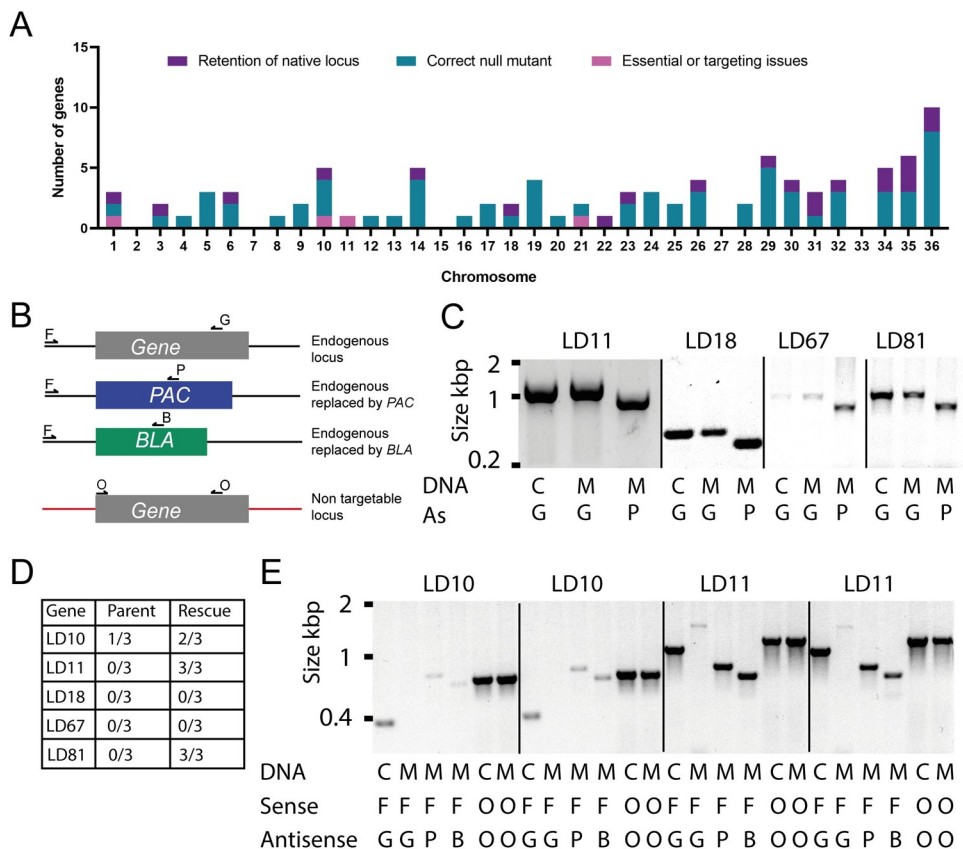

**Fig 2. The failure to obtain some null mutants is unlikely to be due to aneuploidy or inability to correctly target the gene.** (**A**) Graph showing the success of gene targeting on individual chromosomes. The number of successfully targeted genes is shown in teal, failure to recover viable promastigotes in pink, and dual drug resistant parasites retaining a copy of the native loci in purple. Chromosomes 26 and 31 are present in multiple copies per haploid genome. (**B**) Schematic of the diagnostic PCRs used to genotype parasites. (**C**) Diagnostic PCRs demonstrating the expected monoallelic targeting of genes with just the puromycin-selection cassette for genes that had failed to recover doubly-drug resistant parasites. The indicated genomic DNAs are: wild-type control, C or puromycin-resistant mutant parasites, M, which were used as PCR templates using a shared sense primer (F) and the indicated antisense primer as shown in the schematic. (**D**) Table showing the frequency of being able to recover doubly drug resistant parasites in the presence (Rescue) or absence (Parent) of a non-targetable version of the named genes. (**E**) Confirmation of correct integration of drug resistance markers into the *LdBPK_292250.1* (LD10) and *LdBPK_211610.1* (LD11) loci only in the presence of a non-targetable copy of the gene under constitutive expression. Genotyping for LD10 and LD11 is shown for two independently selected populations.

modulate gene copy number may be used by the parasite as a mechanism to regulate gene expression, and it is possible to observe this plasticity experimentally in classical gene knockout studies of genes that have been chemically validated to be essential in this parasite [28,29]. In our experiments, we observed that targeting attempts for 17 genes on at least one occasion

**Table 1. A list of genes with evidence of *in vitro* essentiality from a reverse genetic screen.** Genes are listed with their systematic "LD" number, accession identifier, description and protein architectural class information as assigned from SignalP, HMMER and GPI-pred.

| Systemic ID | Gene ID | Description | Protein class |
|---|---|---|---|
| LD11 | *LdBPK_211610.1* | Hypothetical protein | Secreted |
| LD18 | *LdBPK_100590.1* | Hypothetical protein | Secreted |
| LD67 | *LdBPK_111030.1* | Hypothetical protein | Secreted |
| LD81 | *LdBPK_111030.1* | Pseudouridylate synthase-like protein | Other |

resulted in doubly-drug resistant parasites that nevertheless showed evidence of retention of the endogenous gene (S1 Table). This may indicate that these genes are essential for promastigote growth and so we investigated the gene *LdBPK_292250.1* (LD10) which encodes a putative component of the ergosterol biosynthesis pathway in more detail. In three independent attempts to generate a null mutant for LD10 in the parental strain, doubly-drug resistant parasites were recovered, but at a lower frequency (one out of three attempts) compared to knockouts attempted in the presence of the "rescue" construct (two out of three attempts, Fig 2D). Indeed, it was only in the presence of the rescue construct that correct targeting of both LD10 alleles was observed in both populations (Fig 2E), confirming that C8 sterol isomerase is an essential gene for the viability of *L. donovani* promastigotes *in vitro*. We conclude that a further 17 genes may be required for *L. donovani* promastigote growth *in vitro* but the plasticity of the parasite genome makes further detailed investigation of this class of genes challenging.

## Gene deletion mutants had little to no impact on parasite fitness *in vitro* or *in vivo*

Of the PCR-confirmed null mutant lines selected in these experiments, almost all exhibited no overt phenotype, having very similar rates of *in vitro* growth, morphology and motility compared to the parental line (Fig 3A). There were, however, a few notable exceptions. Parasites targeted at the *LdBPK_061160* (LD54) and *LdBPK_190920* (LD09) loci exhibited a remarkable increased adhesion to the CellBIND-treated polystyrene plastic used to make the tissue culture flasks without affecting growth rate. We have subsequently shown that the *LdBPK_061160* gene encodes a component of the *L. donovani* GPI-anchor biosynthesis pathway because

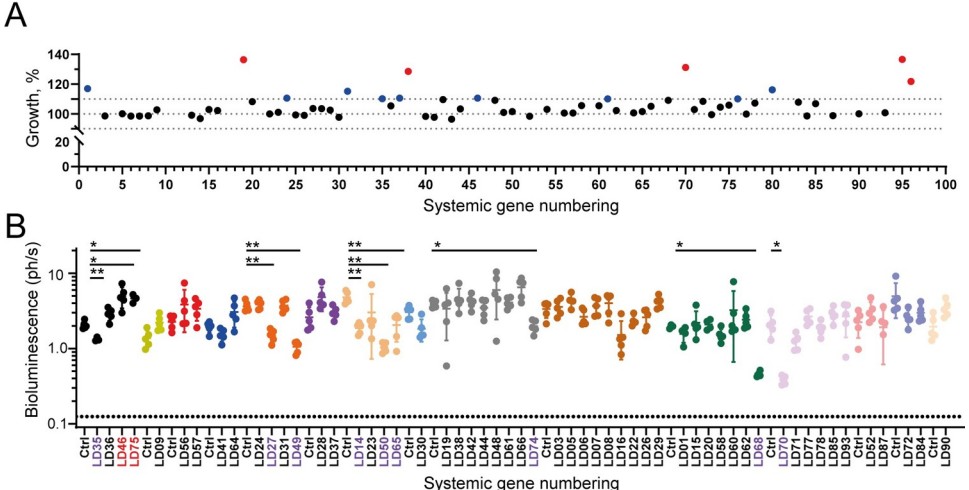

**Fig 3. Most *L. donovani* mutants lacking cell surface or secreted proteins did not affect *in vitro* growth or the ability to infect mammalian hosts.** (**A**) The growth rates of PCR-confirmed gene-targeted *L. donovani* mutants were calculated using promastigotes in logarithmic growth, and were normalised to that of an age-matched parental control from a single culture maintained in logarithmic growth. Black dots represent a < 10% difference in growth rate, blue dots between 10 and 20% difference and red greater than 20% difference compared to the parental line. (**B**) *L. donovani* parasites with targeted mutations in cell surface and secreted proteins were able to establish infections in a mammalian host. Each gene-targeted parasite line was used to infect groups of five female BALB/c mice and the infection monitored by bioluminescent imaging. Colours represent cohorts of animals that were infected with 1x10$^8$ stationary phase promastigotes and liver parasitemia quantified at two weeks post infection using bioluminescence; data points represent mean ± s.d.; $n \geq 3$. Significance was assessed using a Brown-Forsythe and Welch ANOVA test with significance p < .05 (*) or $p < .005$ (**). Systematic numbers have been coloured red for mutants that displayed exacerbated infections and purple for mutants with attenuated infections. Dotted line represents the mean background bioluminescence in uninfected mice. The units of bioluminescence are reported as 1 x 10$^6$ photons per second.

targeting this gene resulted in the loss of major classes of GPI-anchored surface proteins including the GP63 protease and lipophosphoglycans (Roberts *et al.* submitted). Most gene-deficient parasites had no overt effect on *in vitro* growth rates but where differences to the parental line were observed, there were increases rather than decreases in growth rate (Fig 3A). Nine mutants were observed to have an up to 10% increase in growth rate compared to the parental line and a further five had doubling times greater than 10% more than the parental controls (Fig 3A). Mutants with a greater than 10% increase in doubling time included targeted genes encoding a putative dihydrolipoamide acetyltransferase precursor protein, a component of the pyruvate dehydrogenase complex (*LdBPK_210610.1*, LD70), a subtilisin like serine peptidase (*LdBPK_130940.1*, LD19), and an uncharacterised protein (*LdBPK_290220.1* LD95). In summary, parasites with targeted disruption of genes encoding cell surface and secreted proteins generally had little impact on parasite viability *in vitro*, possibly because these gene products do not exhibit cell autonomous phenotypes.

We reasoned that cell surface and secreted proteins generally have roles in host-parasite interactions that would only become apparent during host infection. To identify genes encoding cell surface and secreted proteins that were essentially required for host infection and could therefore make good vaccine candidates, we infected mice with a set (55/68) of the mutant parasites and longitudinally quantified the infection using bioluminescent imaging. We chose to infect mice with each mutant individually rather than as a pool [18,30] because of the possibility that non-cell autonomous rescue effects would confound our analysis. To ensure that the mutants tested spent equivalent times in the stationary phase to differentiate into the infective metacyclic forms, seeding densities were adjusted for those with significantly longer doubling times and groups of five BALB/c mice were challenged using an established bioluminescent imaging model [19]. We quantified the infections in the liver and a region encompassing both the spleen and inguinal lymph node. Twelve of the mutants were found to have significantly altered liver burdens compared to the parental control measured at two weeks post infection with ten showing attenuated infections and two displaying exacerbated infections (Figs 3B and S2). Infection with the *LdBPK_210610.1* (LD70) mutant, previously identified as having an increased doubling time in comparison with the control also displayed an attenuated infection profile demonstrating the importance of this protein *in vitro* and *in vivo*. As none of the mutants assessed were completely incapable of establishing infections *in vivo*, we did not pursue these further as vaccine candidates.

## Vaccinating mice with recombinant protein antigens of genes essential for proliferation *in vitro*, significantly reduces splenic parasite burdens *in vivo*

To assess whether the proteins encoded by the genes essential for *in vitro* parasite viability could elicit immunity in the context of a subunit vaccine, we first attempted to express the four candidates (LD11, 18, 67, and 81) as secreted recombinant proteins. Further to these, another protein LD04 was included because we had initially believed this gene to be essential based on failed knockout attempts; however, an error in a targeting primer was discovered during the review process. We initially selected a mammalian expression system based on human HEK293 cells that has been successful in producing libraries of extracellular proteins from other parasites [31–34]. We observed good levels of expression for LD04 and LD11 which were then purified from spent tissue culture supernatant (Fig 4A). No expression was observed for LD67 or LD81, including when a different cell line from the *Trichoplusia ni* cabbage looper moth was used. Expression of LD18 was minimal and had an increased electrophoretic mobility indicating proteolytic cleavage during expression, and so was not tested as a subunit vaccine.

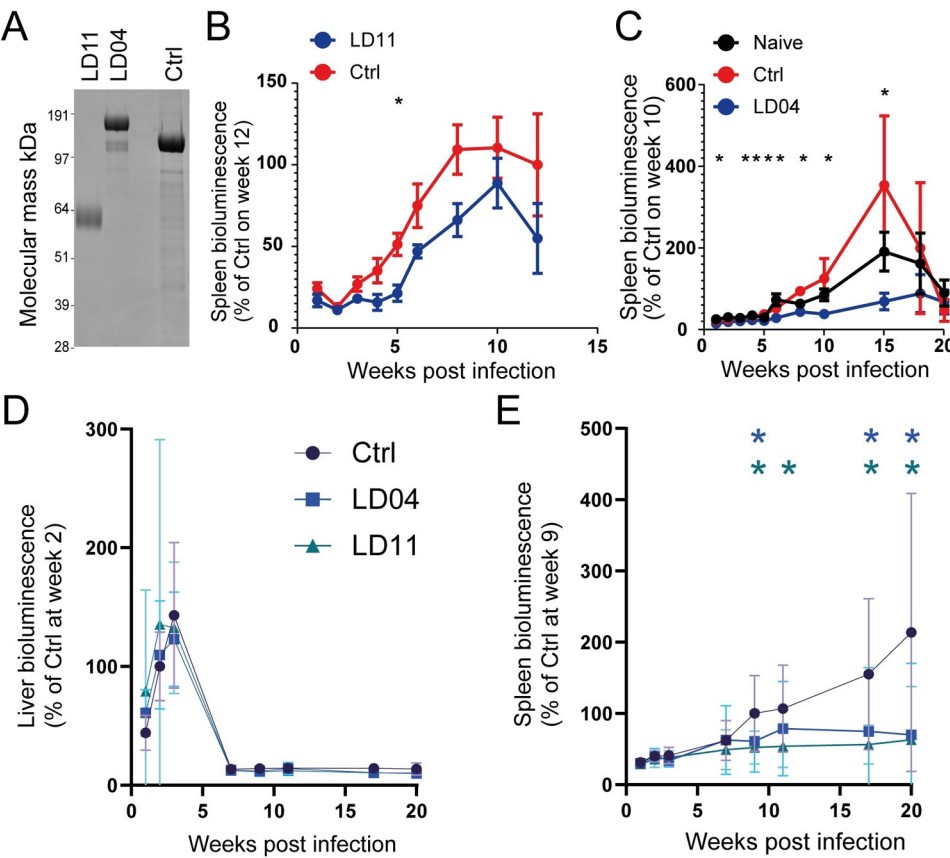

**Fig 4. Candidate antigens LD04 (LdBPK_290790.1) and LD11 (LdBPK_211610.1) elicit protective immune responses against uncontrolled parasitemia in the spleens of infected mice.** (**A**) *L. donovani* vaccine candidates LD04, LD11, and a control protein were expressed as secreted recombinant proteins in HEK293 cells, purified, and then resolved under reducing conditions by SDS-PAGE and stained with Coomassie Brilliant Blue. (**B**) Mice immunised with LD11 adjuvanted in alum display reduced parasitemia in the spleens of infected animals (*n* = 5). Significance was assessed with a *t*-test (*p* < 0.05) between mice immunised with LD11 or a control protein. (**C**) Immunisation with recombinant LD04 adjuvanted in QuilA also demonstrated significant protection against parasitaemia in the spleens of infected mice (*n* = 5). (**D**) No significant reductions in the parasite burdens were observed in the liver of larger cohorts of mice immunised with LD04 or LD11; however, significant reductions in splenic parasite burdens (**E**) were observed (Number of mice in each group was: Ctrl = 15, LD04 = 15 and LD11 = 10). Statistical significance was assessed in all cases using a two-tailed *t*-test at each time point comparing immunisations with either LD04 or LD11 with the control (Ctrl) antigen (* *p* < .05). Data points in panels B-E represent mean ± s.d.

Using a protein-in-adjuvant formulation, we first immunised a group of mice with LD11 in alum and challenged them with 1 x 10⁸ stationary phase promastigotes and quantified the parasite burdens using bioluminescent imaging. Immunised mice had reduced bioluminescent signals in infected spleens indicating reduced levels of parasitemia (Fig 4B). Previous work from our laboratory [33] has demonstrated that altering antibody isotype balances towards immune-effector-recruiting IgG2-isotypes by switching the adjuvant can generate better anti-parasite protective responses [33]. We therefore switched the adjuvant to QuilA for the remaining experiments and immunisation with LD04 elicited significant protection against parasite burdens in the spleens of immunised mice compared to the control immunisation (Fig 4C). Encouraged by these findings, we vaccinated larger group sizes of fifteen animals again using QuilA as the adjuvant. No protection was observed in the initial infection that is localised to the livers of infected animals (Fig 4D); however, immunisation with either LD04 or LD11 was capable of eliciting significant levels of protection against the development of

parasite burdens in the spleens of infected mice (Fig 4E). Although these antigens were unable to elicit complete protection, it is comparable (a 2 to 3-fold reduction) to the result observed with recombinantly expressed HASPB [35]. Taken together, these results suggest that using systematic CRISPR genome editing is a productive approach to identify vaccine candidates for *Leishmania donovani*.

## Discussion

Today, there are over a billion people living in regions endemic for leishmaniasis and therefore at risk of infection; this, together with the lack of an effective human vaccine means leishmaniasis continues to be a global health problem. Proteins exposed to vaccine-elicited antibodies such as those that are secreted or displayed on the surface of the parasite are a promising class of vaccine target, and despite the fact that *Leishmania* spp. parasites are predominately intracellular, there is evidence that this class of proteins can elicit protective immune responses. Dogs can be vaccinated from the visceral disease by immunising with a non-defined mixture of secreted extracts from *L. infantum* promastigote culture by eliciting protective humoral immunity [15,16], providing encouragement that a vaccine for human leishmaniasis could be developed. Modern human vaccines, however, are chemically defined and so a major challenge would be to identify the active molecule/s within this complex mixture. Reverse vaccinology offers one such approach to identify defined vaccine targets for parasitic diseases and this has already had some success for malaria [36–38] and *Trypanosoma vivax* [33]. In terms of the protective immunological mechanisms elicited by a vaccine then antibodies to parasite cell surface proteins could conceivably work by recruiting immune effectors such as complement [33,38] or alternatively by binding and interfering with the usual function of the gene product as is the case for the malaria blood stage vaccine candidate RH5 [36,37,39]. Where the mechanism of vaccine action is to neutralise the normal function of the protein, identifying those genes that are essential for the viability of the parasite in culture, or required for productive infection of the host, would help downselect these candidates from the several thousand encoded in the genome.

In our study, we have exploited recent advances in *Leishmania* genetics [17,18,24,30] and *Leishmania* infection models [19] to target 92 genes predicted to encode extracellular or secreted proteins from *L. donovani* and then assess their role in virulence *in vivo* and importance to cellular growth *in vitro*. Of the 92 genes selected for targeting, it was possible to recover doubly-drug resistant parasites without evidence of the endogenous gene being retained for 68, giving a targeting rate of 74%. The remaining genes either failed to recover any viable parasites or demonstrated the retention of endogenous alleles whilst under dual drug selection. *Leishmania* spp. parasites are known to be able to readily rearrange their genome under strong selection pressure, a phenomenon observed when attempting to select null mutants for chemically validated drug targets [28,29]. This suggests that the 17 genes where doubly-drug resistant parasites were recovered but where the endogenous gene was retained might also be considered as being essential for parasite viability, thereby warranting further investigation. When compared to similar high throughput reverse genetic screens studying proteins localised to the *Leishmania* flagellum [18], or encoding protein kinases [30], our success rate was closer to those targeting protein kinases. This appears to indicate a greater dispensability of flagellum proteins in the parasite than the kinases, many of which are known to be essential for viability [3,30].

One of the major aims of our study was to identify *L. donovani* genes that encoded antibody-accessible proteins that were important for infecting a mammalian host, which would provide important criteria to promote the protein as a subunit vaccine candidate. Of the

mutants for which we were able to readily generate gene deletions, we surprisingly found that none were essentially required to establish an infection in our murine infection model [19]. For those that did show significant differences compared to the parental control, they were nevertheless still ultimately able to establish an infection. These observations are starkly different to those made in another parasite, *Plasmodium berghei*, where 45% of targeted genes were essential for blood stage growth of the parasite *in vivo*, with a further 18% required for normal rates of parasite multiplication [40]. The research in *P. berghei* also noted that proteins predicted to be at the host-parasite interface were more likely to be redundant, which could possibly explain why the secreted and cell surface proteins we have targeted in *Leishmania donovani* were not essential for infection and likely reflect the evolutionary response of the parasite to the host adaptive immune response. A similar systematic gene targeting study in another single-celled eukaryote—*S. cerevisiae*—indicated that up to 80% of the genes are dispensable during *in vitro* growth [41], which is more in line with the results we have obtained here. The similarities between these studies is that selection is done on a rich growth medium *in vitro*; this supportive environment could provide a metabolic rescue facilitating the generation of the null mutants [42,43].

The observation that vaccination with LD04 or LD11 can provide a significant level of protection against developing an uncontrolled splenic infection demonstrates their promise as potential vaccine candidates for visceral leishmaniasis. Efficacy may be improved by further optimising the format of delivery including the co-administered adjuvant with the ultimate aim of improving its efficacy against both the liver and splenic parasite burdens. More detailed studies, however, would be needed to assess if this protection would be replicated in canine and human infections. Notably, the levels of protection offered by these antigens are of a similar magnitude as another recombinantly expressed antigen HASPB [35], that is secreted by the parasite [44]. It should also be possible to identify new candidate antigens for vaccine development against leishmaniasis through improved information on the subcellular localisation of proteins within the parasite and through larger scale genetic screens. In summary, this approach provides a complementary strategy to identify candidate vaccine antigens that bypasses the technical bottleneck of antigen production for vaccines targeting *Leishmania* spp. parasites and contributes to the ultimate goal of developing an effective vaccine against this deadly infectious disease.

## Materials and methods

### Ethics statement

All animal experiments were performed under United Kingdom Home Office regulations (licence numbers P98FFE489 and PD3DA8D1F) and European directive 2010/63/EU. Research was ethically approved by the Sanger Institute Animal Welfare and Ethical Review Board. The animals used were six to eight week-old female *Mus musculus* strain BALB/c obtained from a breeding colony at the Wellcome Sanger Institute. Mice were maintained on a 12-hour light/dark cycle at a temperature of 19 to 24°C and humidity between 40 to 65%.

### Mouse infections with *L. donovani* and bioluminescent imaging

Parasites seeded at $1 \times 10^6$ per mL were grown for seven days and harvested by centrifugation at 2000 x *g* for 10 min. Cell pellets were washed twice with phosphate buffered saline (PBS) and pelleted by centrifugation. The parasites were resuspended in DMEM media, counted, and the concentration of motile, flagellated parasites adjusted to $5 \times 10^8$ parasites per mL. Groups were inoculated intravenously and infections monitored by *in vivo* bioluminescence as described [19]. Briefly, D-luciferin (BioVision, Inc.) was dissolved in PBS and administered

into the peritoneal cavity of animals at 300 mg/kg. Injected animals were anaesthetised using 3% gaseous isoflurane and imaged using an In Vivo Imaging System (IVIS Spectrum, Perki-nElmer). Bioluminescence acquisitions were performed ten minutes after luciferin injection, and data were analysed using the Living Image software version 4.7.4, with bioluminescence signal converted to radiance (photons/sec/cm2/sr) using a fixed normalised scale on IVIS images of animals. Bioluminescence of the liver and spleen were quantified by defining these organs as regions of interest and analysed using GraphPad Prism 8 version 8.3.0. Images were normalised to a fixed scale with radiance limits set at $3 \times 10^3$ (minimum) and $1.5 \times 10^5$ (maximum). An average background bioluminescence measurement was determined by luciferin administration in five female BALB/c mice measuring bioluminescence for areas corresponding to the liver and spleen; where appropriate, this value is indicated as a dotted line on bioluminescence plots. The Brown-Forsythe and Welch ANOVA tests (GraphPad Prism 8) with Dunnett T3 correction for multiple testing were used to determine the significance of measured signals within each cohort.

## *L. donovani* culture and transgenesis

*Leishmania* promastigotes were cultured in RPMI-1640 supplemented with 20% (v/v) heat-inactivated foetal calf serum, 100 μM adenine, 20 mM 2-(N-Morpholino) ethanesulfonic acid hydrate, 5 μM hemin, 3 μM 6-biopterin and 1 μM biotin [45]. Electroporation of plasmid DNA or PCR products were performed as previously described [17]. A parental *L. donovani* strain expressing both firefly luciferase and mCherry [19] was electroporated with plasmid pTB007 and selected to constitutively express *Sp*Cas9 and T7 RNA pol. Gene-deficient parasites were selected by electroporation a mixture of PCR products encoding sgRNA templates and drug resistance markers. Transgenic parasites were selected by the addition of puromycin, blasticidin, nourseothricin or hygromycin B at 15, 25, 100 and 50 μg mL$^{-1}$ respectively, 24 hours post electroporation. Doubling times of the parental and transgenic parasites were calculated from populations of parasites maintained in continuous log phase growth over a minimum of three successive subcultures. Growth rates were obtained by fitting the data directly to an exponential growth model fitted using the least squares method with GraphPad Prism 8.

## CRISPR/Cas9 gene targeting in *L. donovani*

Primers used to design repair or sgRNA templates (S2 Table) were obtained from LeishGEdit (http://www.leishgedit.net/Home.html) and compared to the genome sequence of the parental transgenic *L. donovani* strain [19]; these sequences were used if they matched exactly, or were altered as appropriate if polymorphisms were observed. Protospacer adjacent motif sequences were checked for potential off-target activity against the genome of the experimental strain using Cas-OT (S3 Table) [46]. The PCR products used as templates for sgRNAs or repair cassettes were amplified as previously described [17] using Q5 Hot-start polymerase (NEB). The plasmids pTPuro_V1 and pTBlast_V1 provided the templates of the repair cassettes for generating null mutants. Correct targeting of genes was confirmed by the absence of a PCR product using genomic DNA (gDNA) extracted from drug-resistant parasites compared to the parental line by using gene-specific primers that annealed within the 5' UTR (sense) and ORF (anti-sense) (S4 Table). To confirm targeted integration of the genes encoding both drug selection markers, anti-sense or sense primers for blasticidin S deamidase (*BLA*) and puromycin N-acetyltransferase (*PAC*) genes were used with the gene-specific sense primers. These amplifications were carried out using REDTaq polymerase (Sigma) using 200 ng gDNA under the following conditions. Denaturation for 30 s at 98˚C, followed by 30 cycles of 98˚C for 30 s, 63˚C for 30 s and 72˚C for 85 s with a final 5 min extension. PCR products were analysed on 1% agarose gels.

## Phenotype rescue by gene overexpression

NheI and XhoI restriction sites were introduced immediately 5' and 3' of the T7 RNA polymerase sequence of the plasmid PTB008 (a generous gift from Dr Eva Gluenz, University of Glasgow) using the NEB Q5 site directed mutagenesis kit using the primers listed in (S5 Table). The resulting plasmid was digested with NheI and XhoI restriction endonucleases and a new MCS introduced by ligation of the following annealed oligo pair primer 1 CTAGCgcgg ccgcGGATCCTCTAGAc and primer 2 tcgagTCTAGAGGATCCgcggccgcG, yielding the plasmid PTBLE. The entire open reading frames of specific genes were amplified with primers that incorporated these restriction enzyme sites (S5 Table) and the digested PCR products were ligated into the linearized PTBLE plasmid.

## Protein expression and purification and immunisations

Codon optimised ectodomains fused to an N-terminal signal peptide for secretion and a C-terminal biolinker containing a hexahistidine tag were transfected in to HEK293-6E [47], HEK293E [48] or High Five insect cells using PEI MAX. Recombinant ectodomains were purified from supernatants using nickel affinity chromatography (HisTrap GE Healthcare) and eluted with a gradient of imidazole. Purified recombinant proteins were dialysed into PBS using a 3.5 kDa MWCO membrane. Recombinant proteins were co-administered with the adjuvant QuilA (Fig 4C–4E) containing 50 µg of protein per mouse or were mixed with 2% alum (Alhydrogel, InvivoGen) (Fig 4B) at a 1:1 (v/v) ratio and incubated on a roller mixer for 1 h at 22°C. Immunisations were administered via the subcutaneous route and boosted at 2 and 4 weeks after the initial immunisation and allowed to rest for 4 weeks prior to infection with a bioluminescent strain of *Leishmania donovani* as described above. Rat CD200 was used as the control antigen (Fig 4B and 4C) whilst *Pf*TRAP [49] was used as the control (Fig 4D and 4E).

## ELISAs

A total of 1 ng of the recombinant ectodomains were immobilized per well of a 384 well maxisorp plate overnight at 4°C. Non-specific binding was blocked by incubating with 5% (w/v) bovine serum albumin dissolved in phosphate buffered saline, containing 0.05% (v/v) Tween-20 (PBS-T) for 1 hour at room temperature. Serum from immunised mice was collected and serially diluted into PBS-T before incubated with the immobilised proteins for 2 hr at room temperature. Wells were washed with PBS-T three times to remove all unbound antibodies and then incubated with goat anti-mouse IgG conjugated with alkaline phosphatase (1 in 2500 dilution) for a further hour at room temperature. Excess secondary antibody was removed by three washes with PBS-T. Samples were incubated with 4-Nitrophenyl phosphate disodium salt hexahydrate dissolved in 10% diethanolamine, 0.5mM $MgCl_2$, pH9.2 at 1 mg ml$^{-1}$ and analysed by measuring absorbance at 405 nm using a Tecan Spark plate reader. Data was analysed in GraphPad Prism (v8.3.0).

## Western blot analysis

Equal numbers of parasites were harvested by centrifugation at 2000*g* for 10 min. Cell pellets were washed in PBS and re-harvested by centrifugation and resuspended in Laemmli buffer and heat inactivated at 95°C for 10 min. Soluble lysates (5 min at 10,000*g*) were resolved by SDS-PAGE and transferred to a methanol activated nylon membrane by electrotransfer using the NUPAGE transfer buffer and XCell II blot module at 30 V for 1 h. The membrane was blocked in PBS-T containing 5% (w/v) BSA for an hour at room temperature. The membrane

was probed with the horseradish peroxidase conjugated mouse anti-FLAG M2 monoclonal (1 in 5000 dilution). Excess antibody was removed by washing in PBS-T and binding detected using the Pierce ECL western blotting detection kit using Amersham Hyperfilm ECL and developed.

## Supporting information

**S1 Table. Details of the *L. donovani* genes targeted in this study.** Each gene is given a systematic "LD" number for convenience together with its accession number. The number of knock-out attempts and chromosome number for each gene is also noted.
(XLSX)

**S2 Table. List of primers used in the generation of the null mutant library.**
(XLSX)

**S3 Table. Guide RNA off-target predictions in the *L. donovani* luciferase expressing genome using Cas-OT.** Columns list the number of sequences that match the guide RNA sequence with a number of mismatches in the seed region and non-seed region of the protospacer adjacent motifs. A00 is a perfect match, A10 is the number of sequences in the genome that match with a single mutation in the seed region. A05 lists the number of sequences that match the guide sequence with 5 mutations in the more permissive non-seed region. All guides should have a single match for A00, the targeted cut site where the genome has been assembled. Guides highlighted in red do not have a perfect match to the assembled genome strain, but do perfectly match the UTR's identified from the unassembled genome of the experimental strain.
(XLSX)

**S4 Table. List of diagnostic primers used in this study.** LD54 primers were used with the PAC and BLA sense primers. Integration of all others were assessed with the PAC and BLA antisense primers.
(XLSX)

**S5 Table. List of primer sequences used in the repurposing of PTB008 to PTBLE and for constitutive overexpression of select genes.**
(XLSX)

**S1 Fig. Gene-specific products of diagnostic PCRs to determine the success of gene targeting across the library of Ld-LUC-T7-Cas9 parasites.** Individual boxes show diagnostic PCR products amplified from genomic DNA extracted from parasites resistant to both puromycin and blasticidin to confirm correct gene targeting; each gene is indicated using the systematic numbering system. For each gene, the products of four PCRs are shown. Lanes 1 and 2 indicate presence of the native allele in genomic DNA extracted from the parental *L. donovani* strain (lane 1), and targeted doubly-drug resistant parasites (lane 2). Locus-specific targeting with each of the drug resistance genes was demonstrated using primers that were specific to the target locus and either the puromycin (lane 3) or blasticidin (lane 4) resistance cassettes. Note that for gene LD54, the lanes that the PCR products using the puromycin and blasticidin-specific primers were switched. Multiple off-target amplification bands were detected in the LD63 amplifications, and a slower migrating band was observed in the LD80 null mutant. The diagnostic amplification of the correct integration of PAC into the LD90 locus failed on multiple occasions, but we deemed it to be a null mutant as the parasites were resistant to puromycin and the amplification of the endogenous locus showed disruption.
(TIF)

**S2 Fig. Quantification of the liver parasite burden reveals no essential role for 54 *L. dono-vani* genes encoding cell surface and secreted proteins in mammalian host infections.** Parasitemia in groups of five female BALB/c mice that had been challenged with 1 x $10^8$ stationary phase gene-targeted *L. donovani* promastigotes was quantified using bioluminescent imaging at days 0, 3, 8, 16, 22 and 30 days post infection. Separate graphs plot the results from cohorts of mice infected with the indicated mutant parasites. Data points represent means ± s.d.; *n* = 5. (TIF)

## Acknowledgments

We would like to thank Dr Eva Gluenz and Tom Beneke for providing the plasmids and Mary Wilson for providing the parental luciferase expressing *L. donovani* strain used in this study.

## Author Contributions

**Conceptualization:** Adam J. Roberts, Gavin J. Wright.

**Data curation:** Adam J. Roberts.

**Formal analysis:** Adam J. Roberts, Susanne U. Franssen, James A. Cotton.

**Funding acquisition:** Gavin J. Wright.

**Investigation:** Adam J. Roberts, Han B. Ong, Simon Clare, Cordelia Brandt, Katherine Harcourt, Nicole Müller-Sienerth.

**Methodology:** Adam J. Roberts, Han B. Ong, Simon Clare.

**Project administration:** Gavin J. Wright.

**Writing – original draft:** Adam J. Roberts, Gavin J. Wright.

**Writing – review & editing:** Adam J. Roberts, Gavin J. Wright.

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
