## [Decision Letter · Decision Letter 0]

18 Oct 2021

Dear Dr. Wright,

Thank you very much for submitting your manuscript "Systematic identification of genes encoding cell surface and secreted proteins that are essential for in vitro growth and infection in Leishmania donovani." for consideration at PLOS Pathogens. As with all papers reviewed by the journal, your manuscript was reviewed by members of the editorial board and by several independent reviewers. The reviewers appreciated the attention to an important topic. Based on the reviews, we are likely to accept this manuscript for publication, providing that you modify the manuscript according to the review recommendations. The revised manuscript should address each of the minor issues that were raised.

Sincerely,

Dan Zilberstein

Guest Editor

PLOS Pathogens

David Sacks

Section Editor

PLOS Pathogens

Kasturi Haldar

Editor-in-Chief

PLOS Pathogens

orcid.org/0000-0001-5065-158X

Michael Malim

Editor-in-Chief

PLOS Pathogens

orcid.org/0000-0002-7699-2064

The revised manuscript to address each of the minor issues that were raised by the reviewers.

Reviewer Comments (if any, and for reference):

Reviewer's Responses to Questions

**Part I - Summary**

Reviewer #1: The study by Roberts et el represents a considerable body of work to target 92 genes encoding secreted or membrane bound proteins, with the rational that antibodies against some of these secreted proteins may impair parasite survival and thus may represent vaccine targets. The rational for this approach is interesting, testable as conducted in this study and has the study therefore has strong merit. One consideration however is that although the Leishmania protein may be secreted, it may not make it outside the phagolysosome or the infected macrophage and thus would still not be accessible to extracellular antibodies. Nevertheless, I believe the determining the role of this interesting set of leishmania genes in parasite survival in mice is an important study. What is perhaps unexpected was how few of these 92 genes are actually needed for parasite survival in the mouse (none!) although some led to attenuation. Targeting these gene products with antibodies was an interesting approach.

Reviewer #2: In spite of the flawless execution of the study, the fact remains that the majority of the work gave inconclusive results. None of the preselected genes have a conditional effect during the mammalian stage of the parasite. There are some marginal growth effects, but no critical impact on the infectivity/pathogenicity of the parasites.

Immunization of mice with two selected proteins (LD04, LD11) did not protect against an initial burst of parasite burden in the liver, but prevented the increase of parasite burden in the spleens of the animals, indicating a protective effect against chronic, visceral infection. However, the use of a Students t-test with a sample size of n=5 is not ideal; a non-parametric test (ANOVA, U-test) might be more appropriate. The results should also be shown as a scatter graph to give an idea of the variances.

Reviewer #3: In this manuscript, Roberts et al describe a study that uses CRISPR/Cas9 to systematically target 92 genes from Leishmania donovani encoding proteins that are predicted to be secreted or anchored to the external cell membrane. Only 5 genes appeared to be essential for in vitro growth of promastigotes, although they were also unable to obtain null mutants for another 12. Nine additional genes appeared to be important for virulence in the mammalian host, since null mutants showed attenuated infection in mice. Finally, the proteins encoded by the 5 essential genes were used to vaccinate mice and two showed significant protection against splenic infection. The study is well designed with adequate controls and, in general, the manuscript is well-written and the conclusion (that the two proteins should be further investigated as vaccine candidates) is warranted.

**Part II – Major Issues: Key Experiments Required for Acceptance**

Reviewer #1: There is no major issue with the study other than the obvious; the strategy failed to identify a vaccine target. However, I applaud the effort since there was some rational in the approach and it was very interesting to learn that none of the targeted proteins are needed for survival in the mouse model.

Reviewer #2: Given the difficulties in producing the vaccination proteins, the use of recombinant virus vaccination should be considered to broaden the last part of the study.

Reviewer #3: There are no major issues that need to be addressed.

**Part III – Minor Issues: Editorial and Data Presentation Modifications**

Reviewer #1: I had a number of comments which I would place in the Minor issue category as listed here.

Why were the genes needed for promastigote growth selected for vaccine targets, I would have thought the genes needed for optimum survival in the mouse would have been equally good targets? Examples Ld68, LD70, Ld50

With respect to CL parasites protecting against VL parasites (line74-75), a newer reference should now also be included (https://doi.org/10.1038/s42003-021-02446-x)

The first results paragraph describes the development of the transgenic line, how it was unclear from the description how the luciferase gene was expressed in this line.

Line 179, please provide some explanation how genes in chrom 26 and 31 were targeted using 2 selectable markers for trisomy and tetrasomy chromosomes. Did the same marker enter 2 chromosomes. If so, presumable a single marker could have been used for all the targeted genes and this could have been verified by PCR such as in Table S1.

The description of proliferation in Fig 3A are specific to promastigotes; not parasites since parasites could include amastigotes.

Line 274; it would have been better to isolate equivalent numbers of metacyclic promastigotes from each culture to perform the BALB/c mouse infections. Mutants with lower levels of infection could have had fewer metacyclics, particularly mutants in membrane proteins.

Please see attachment for further comments.

Reviewer #2: Everything was concisely presented.

Reviewer #3: Lines 75 and 84: there are several instances throughout the manuscript where Leishmania is not italicized.

Line 115: What strain/isolate of L. donovani was used. Isolates from different geographical location show considerable genetic variation and it is important to know precisely which one was used.

Line 166: remove the space before the period.

Line 181: While the identify of all 92 genes (or at least their orthologues in the LdBPK282A1 reference strain) are identified in supplementary Table S2, it might be useful for the reader if a table were added to the main text that summarized at least the 5 essential genes.

Figure 3: Replace “systemic” with “systematic”

Lines 402-406: It is not clear what is the point of the discussion about P. berghei genes.

PLOS authors have the option to publish the peer review history of their article (what does this mean?). If published, this will include your full peer review and any attached files.

Reviewer #1: No

Reviewer #2: No

Reviewer #3: **Yes: **Peter J Myler

Figure Files:

Data Requirements:

Reproducibility:

References:

---

## [Editor Report · Decision Letter 1]

11 Feb 2022

Dear Dr. Wright,

We are pleased to inform you that your manuscript 'Systematic identification of genes encoding cell surface and secreted proteins that are essential for in vitro growth and infection in Leishmania donovani.' has been provisionally accepted for publication in PLOS Pathogens.

Best regards,

David Sacks

Section Editor

PLOS Pathogens

David Sacks

Section Editor

PLOS Pathogens

Kasturi Haldar

Editor-in-Chief

PLOS Pathogens

orcid.org/0000-0001-5065-158X

Michael Malim

Editor-in-Chief

PLOS Pathogens

orcid.org/0000-0002-7699-2064
---

## [Editor Report · Acceptance letter]

21 Feb 2022

Dear Dr. Wright,

We are delighted to inform you that your manuscript, "Systematic identification of genes encoding cell surface and secreted proteins that are essential for in vitro growth and infection in Leishmania donovani.," has been formally accepted for publication in PLOS Pathogens.

Best regards,

Kasturi Haldar

Editor-in-Chief

PLOS Pathogens

orcid.org/0000-0001-5065-158X

Michael Malim

Editor-in-Chief

PLOS Pathogens

orcid.org/0000-0002-7699-2064